# Prevalence of Orthodontic Malocclusions in Healthy Children and Adolescents: A Systematic Review

**DOI:** 10.3390/ijerph19127446

**Published:** 2022-06-17

**Authors:** Lutgart De Ridder, Antonia Aleksieva, Guy Willems, Dominique Declerck, Maria Cadenas de Llano-Pérula

**Affiliations:** 1Department of Oral Health Sciences, Orthodontics KU Leuven & Dentistry, University Hospitals Leuven, 3000 Leuven, Belgium; antonia.aleksieva92@gmail.com (A.A.); guy.willems@kuleuven.be (G.W.); maria.cadenasdellanoperula@kuleuven.be (M.C.d.L.-P.); 2Department of Oral Health Sciences, Research Group Population Studies in Oral Health and Pediatric Dentistry & Special Care, University Hospitals Leuven, 3000 Leuven, Belgium; dominique.declerck@kuleuven.be

**Keywords:** prevalence, malocclusion, orthodontic features, children, adolescents

## Abstract

The purpose of this study was to systematically review the literature regarding the prevalence of malocclusion and different orthodontic features in children and adolescents. Methods: The digital databases PubMed, Cochrane, Embase, Open Grey, and Web of Science were searched from inception to November 2021. Epidemiological studies, randomized controlled trials, clinical trials, and comparative studies involving subjects ≤ 18 years old and focusing on the prevalence of malocclusion and different orthodontic features were selected. Articles written in English, Dutch, French, German, Spanish, and Portuguese were included. Three authors independently assessed the eligibility, extracted the data from, and ascertained the quality of the studies. Since all of the included articles were non-randomized, the MINORS tool was used to score the risk of bias. Results: The initial electronic database search identified a total of 6775 articles. After the removal of duplicates, 4646 articles were screened using the title and abstract. A total of 415 full-text articles were assessed, and 123 articles were finally included for qualitative analysis. The range of prevalence of Angle Class I, Class II, and Class III malocclusion was very large, with a mean prevalence of 51.9% (SD 20.7), 23.8% (SD 14.6), and 6.5% (SD 6.5), respectively. As for the prevalence of overjet, reversed overjet, overbite, and open bite, no means were calculated due to the large variation in the definitions, measurements, methodologies, and cut-off points among the studies. The prevalence of anterior crossbite, posterior crossbite, and crossbite with functional shift were 7.8% (SD 6.5), 9.0% (SD 7.34), and 12.2% (SD 7.8), respectively. The prevalence of hypodontia and hyperdontia were reported to be 6.8% (SD 4.2) and 1.8% (SD 1.3), respectively. For impacted teeth, ectopic eruption, and transposition, means of 4.9% (SD 3.7), 5.4% (SD 3.8), and 0.5% (SD 0.5) were found, respectively. Conclusions: There is an urgent need to clearly define orthodontic features and malocclusion traits as well as to reach consensus on the protocols used to quantify them. The large variety in methodological approaches found in the literature makes the data regarding prevalence of malocclusion unreliable.

## 1. Background

In the 1890s, E. Angle defined normal dental occlusion as follows “the upper and lower molars should be related so that the mesio-buccal cusp of the upper molars occludes in the buccal groove of the lower molars and with the teeth arranged in a smoothly curving line of occlusion” and classified malocclusion in four classes (normal occlusion, Class I, Class II and Class III malocclusion) based on the relationship between the upper and lower first molars.

Furthermore, the World Dental Federation (FDI) states that “malocclusion may affect oral health by increasing the prevalence of dental caries, periodontitis, risk of trauma and difficulties in masticating, swallowing, breathing and speaking” and that “orthodontic care has evolved to become an integral part of dentistry helping to prevent oral disease and improve quality of life” [1].

In this context, information regarding the prevalence of malocclusion and the overall need for orthodontic treatment is essential to provide objective information to healthcare stakeholders, to allow for the allocation of healthcare resources based on objective epidemiological data. This information is also crucial for the training of dental and orthodontic healthcare professionals and for the rational planning of all aspects of orthodontic care [2,3].

Despite these facts, large and representative epidemiological studies regarding orthodontic features are hard to find. Proffit et al. argued that the lack of consensus among researchers regarding how much deviation from the ideal should be accepted as normal to be a possible explanation for this [4].

The Third National Health and Nutrition Examination Survey (NHANES III), which was performed in the United States from 1989 to 1994, collected data on the prevalence of malocclusion. A 30% prevalence of Angle “normal occlusion” and a 50–55%, 15%, and <1% prevalence of Angle Class I, II, and III malocclusion were reported, respectively. However, the molar relationship was not examined directly, but rather derived from the overjet measurements, which were claimed to be evaluated more precisely [4,5]. A systematic review on the prevalence of malocclusion in Chinese schoolchildren found 30.1%, 9.9% and 4.8% Angle Class I, II, and III malocclusion, respectively. They also reported deep bite to be the most common malocclusion trait, observed in 16.7% of the sample [6]. Another systematic review reported the prevalence of malocclusion in Iranian children to be 54.6%, 24.7%, and 6.0% for Angle Class I, II, and III, respectively [7]. Knowledge of the prevalence of extensive orthodontic features such as oral clefts, craniofacial syndromes, oligodontia and others is also important in terms of burden of care. According to the World Health Organization (1998), lip, alveolus, and/or palate clefts affect between 1 out of 500 (0.2%) and 1 out of 700 (0.1%) live births in Europe [8].

The aims of this article are firstly to systematically review the existing literature regarding the prevalence of malocclusion and different orthodontic features in children and adolescents and secondly to identify possible inconsistencies in definitions and measurement protocols.

## 2. Materials and Methods

### 2.1. Protocol and Registration

The protocol of this systematic review was drafted prior to data collection, and the results are reported according to the PRISMA guidelines (Preferred Reporting Items of Systematic Reviews and Meta-analysis) [9]. The protocol was registered in the international prospective register of systematic reviews (PROSPERO) under protocol registration number CRD42018086464.

### 2.2. Search Strategy

The digital databases PubMed, Cochrane, Embase, Open Grey, and Web of Science were searched from inception to the 18th of November 2021 by two authors (L.D.R. and M.C.d.L.-P.). Specific search strings were developed per database, which were validated by an expert librarian from the Biomedical Library of KU Leuven, Belgium, and are available as Appendix A. Although the search terms ‘cleft lip and/or palate’ and ‘craniofacial syndromes’ were initially included in the search, articles focusing on these patients were kept separately since they are out of the scope of the present review.

### 2.3. Eligibility Criteria

The inclusion criteria were defined following the PIO format as follows:

Patients: Healthy Subjects ≤ 18 years of age.

Intervention: Assessment of malocclusion and/or dental characteristics.

Outcome: Prevalence and/or incidence of dental malocclusion and dental anomalies, 

Epidemiological surveys, randomized controlled trials, clinical trials, and comparative studies were considered. Papers in English, Dutch, French, German, Spanish, and Portuguese were included. 

Case reports, conference proceedings, letters to the editors, and unpublished studies as well as studies in other languages than the ones mentioned above and studies involving subjects who had undergone orthodontic treatment were excluded.

### 2.4. Study Selection

Publications retrieved from the different databases were imported into a reference manager (Mendeley Ltd., London, UK), and duplicates were removed. In a first phase, the titles and abstracts of all of the retrieved articles were screened by two reviewers (L.D.R. and M.C.d.L.-P.). Afterwards, the full texts of the remaining articles were read by three observers (L.D.R., M.C.d.L.-P. and A.A.), who also performed data extraction and scored the risk of bias. Any disagreements that occurred during the first and second selection phase were discussed until consensus was reached.

### 2.5. Data Collection and Analysis

The following information was extracted from the included studies: the study characteristics (author, publication year, study design, country in which the study was performed, and number of participants), the sample characteristics (type of participant, age, and gender), the type of examination, and a description and assessment of the studied parameters (Angle Class I, Angle Class II, Angle Class II,1, Angle Class II,2, Angle Class III, overjet, reversed overjet, open bite, crowding, spacing, crossbite, scissor bite, forced bite (crossbite with lateral or frontal shift), hypodontia, supernumerary teeth, dental anomalies, impacted/retained teeth, ectopic teeth eruption, tooth transposition, and oral habits).

These data were compiled into datasets in Excel files, and—if possible—the weighted means and weighted standard deviations were calculated to consider the prevalence and its standard deviation relative to the number of subjects in the respective studies. Results were afterwards reported in the sagittal, vertical, and transversal dimension in order to offer a more comprehensive explanation.

### 2.6. Risk of Bias Assessment

The Methodological Index for Non-Randomized Studies (MINORS) from Slim et al., 2003, was used to assess the risk of bias of the included studies [10]. This tool contains 12 items related to comparative studies, the first 8 of which are also applied to non-comparative studies. Each item on the MINORS tool is scored as 0 (not reported), 1 (reported but inadequate), or 2 (reported and adequate), resulting in an ideal total score of 16 for non-comparative studies and 24 for comparative studies.

## 3. Results

The initial electronic database search identified a total of 6775 articles. After the removal of 2129 duplicates, further title and abstract screening as well as an eligibility assessment resulted in the final inclusion of 123 papers for qualitative analysis. Figure 1 shows the PRISMA flow diagram. The characteristics of the studies population and the methods used in the included studies can be found in Table 1 and will be discussed in the following paragraphs. The exact definitions of all orthodontic terms are available at Proffit et al. [4].

### 3.1. Characteristics of the Studied Population

The characteristics of the 123 included articles can be found in Table 1. Most of the studies were performed in a sample of children or schoolchildren (89/123): 9 involved patients and 23 patient records, 1 article included both patients and patient records, and 1 included schoolchildren and patient records. Most of the studies were performed in Europe (42/123), followed by Asia (41/123), America (24/123), Africa (14/123), and Oceania (2/123). X articles did not mention sex distribution. A total of 58 articles found no statistically significant differences in prevalence of malocclusion types between females and males [11,12,13,15,18,21,22,28,29,31,33,35,37,42,44,46,47,49,50,51,52,55,56,57,59,61,67,69,70,72,73,77,79,81,83,85,86,88,94,95,96,98,99,100,106,110,111,113,115,119,122,125,128,129,130,131,132,133].

### 3.2. Methods Used in the Included Studies 

The methods used in the included articles can also be found in Table 1. Clinical examinations (94/123), X-rays (39/123), study casts (20/123), intra- and extra-oral photographs (6/123), and interviews or questionnaires (12/123) were the most frequently used diagnostic methods. To assess malocclusion and orthodontic features, the method of Björk (15/123) or the Angle Classification (15/123), the Index of Orthodontic Treatment Need (16/123), or the Dental Aesthetic index (18/123) were explicitly used. However, the vast majority of the included studies used a non-validated method that was specific to the study.

### 3.3. Prevalence of Malocclusion

#### 3.3.1. Sagittal Occlusion

The terminal plane of the deciduous molar was assessed in 10 of the included studies. A flush terminal plane was found in 41.7 ± 15.2% of the included studies (range 18.2–84.3%.); a distal step was found in 12.4 ± 8.1% (range 0.0–33.6%), and a mesial step in 38.5 ± 10.7% (range 6.0–65.9%).

Regarding the permanent molar, 52 studies reported Angle class occlusion. The mean prevalence for Angle Class I “normal occlusion” was 46.3 ± 27.3% (range 1.7–93.6%); for Class I malocclusion, it was 46.5 ± 17.0% (range 7.4–84.0%); for Class II malocclusion, it was 25.0 ± 13.2% (range 0.8–72.1%); for Class II,1 malocclusion, it was 16.7 ± 12.7% (range 1.7–40.0%); for Class II,2 malocclusion, it was 4.7 ± 2.4% (range 1.4–13.2%); and for Class III malocclusion, it was 7.0 ± 7.9% (range 0.5–39.1%). Large variation was observed in the definitions, measurements, and prevalence of overjet and reverse overjet, which can be found in Table 2.

#### 3.3.2. Vertical Occlusion

The prevalence of overbite and open bite varied considerably, as seen in Table 2.

**Table 2 ijerph-19-07446-t002:** Prevalence of overjet, reversed overjet, overbite, and open bite.

First Author, Year	Subjects	Age Range (Total Sample)	Overjet	Reversed Overjet (Mandibular Overjet)	Overbite	Open Bite	Anterior Open Bite	Posterior Open Bite
	Total Number and Groups if Available	Age Range, and If no Range, Mean Age ± SD						
Abu Alhaija, 2005 [12]	1003	13–15	4–6 mm: 21.7% >6 mm: 3%	1.9%	4–6 mm: 15.9% >6 mm: 1%	4–6 mm: 1.9% >6 mm: 1.0%		
Abumelha, 2018 [13]	526	6–12			deep bite: 21.3%	40.1%		
Alajlan, 2019 [14]	520	7–12	<2 mm: 5% 2–4 mm: 71.2% >4 mm: 14.4% edge–edge: 4.2%	5.2%	2–4 mm: 83.8% >4–7 mm: 11% >8 mm: 5.2%		7.7%	0.6%
al-Emran, 1990 [17]	500	13.5–14.5	5–8.9 mm: 17.2% >9 mm: 1.2%	0–1.9 mm: 2.6% >2 mm: 0.6%	3–4.9 mm: 17.4% >5 mm: 3.6%		0.1–1.9 mm: 3.6% >2 mm: 3%	
Arabiun, 2014 [21]	1338	14–18				1.2%		
Araki, 2017 [22]	420	10–16	>6 mm: 2.4%	<−1 mm: 0.7%	>3 mm: 5.5%	≤4 mm: 0.0%		
Baskaradoss, 2013 [27]	300	11–15	>2 mm: 14%	>2 mm: 2.7%		>1 mm: 3.7%		
Behbehani, 2005 [28]	1299	13–14	0–3.5 mm: 53.2% 4–6 mm: 35% 6.5–9 mm: 6.4% >9 mm: 1.4%	4.0%	2/3–3/3 overlap: 22% >3/3 overlap with gingival contact: 1.7%	3.4%		
Berneburg, 2010 [29]	2015	4–6	0–2.5 mm: 82.2% >2.5 mm: 16.5%	1.3%	0–2 mm: 69.9% >2 mm: 25.5%	4.6%		
Bhardwaj, 2011 [30]	622	16–17	0–2 mm: 73.0% >2 mm: 27.0%	1.1%		1.0%		
Bhayya, 2011 [31]	1000	4–6	0–2 mm: 84.5% 2–4 mm: 11.9% >4 mm: 3.6%		0–2 mm: 81.6% 2–4 mm: 15.7% >4 mm: 2.7%		1.0%	
Bilgic, 2015 [32]	2329	12–16	0–4 mm: 73.5% >4 mm: 25.1%	<0 mm: 10.4%	0–4 mm: 73.5% >4 mm: 18.3%	8.2%		
Bourzgui, 2012 [33]	1000	8–12	0 mm: 5.9% 1–4 mm: 63.9% 4–6 mm: 17.2% >6 mm: 10% Indefinite: 1%	<0 mm: 2%	0 mm: 7.1% 1–4 mm: 65.4% 4–6 mm: 16.6% >6 mm: 7% Indefinite 3.9%	0 mm: 97.1% <3 mm: 1.7% >3 mm: 1.2%		
Calzada Bandomo, 2014 [34]	210	5–11	>9 mm: M: 29.1%–F: 27%		increased (no mm): M: 22.7%–F: 15%	M: 6.4%–F: 13%		
Carvalho, 2011 [36]	1069	5–5 Y11M	>2 mm: 10.5%		>2 mm: 19.7%	7.9%		
Chauhan, 2013 [37]	1188	9–12	0–2 mm: 63.7% >2 mm: 36.3%	≥1 mm: 1.3%			≥1 mm: 0.8%	
Ciuffolo, 2005 [38]	810	11–14	>3 mm: 19.1% >5 mm: 6.5%	negative OJ: 1.1%	>3 mm: 41% >5 mm: 9.6%			
Coetzee, 2000 [39]	214	3–8	mean overjet 2.71 mm	1.9%	deep-3/10 overlap: 18.7% edge to edge: 18.7%		10.3%	
Cosma, 2017 [40]	172	3–6	OJ > 4 mm: 14%		Abnormal OB: 9% (not defined)	11.0%		
Dacosta, 1999 [41]	1028	11–18	<2 mm: F: 20.4%-M: 17.1% 2–4 mm: F: 69.7%-M: 72.1% 5–8 mm: F: 7.5%-M: 7.6% 8–12 mm: F: 0.4%-M: 0.8% >12 mm: F: 0%–M: 0.2%	F: 2%–M: 2.1%	<1/3 overlap: F: 72.4%-M: 66.1% >1/3 overlap but does not exceed middle 1/3 of crown: F: 18.9%-M: 26.0% >overlap middle 1/3 of crown: F: 1.8%–M: 1.5%	F: 4.8%–M: 4.3%		
de Almeida, 2008 [43]	344	3.94 *	>3 mm: 16%		>3 mm: 7%	27.9%		
de Araújo Guimarães, 2018 [44]	390	8–10	≥4 mm: 15.6%				≥2 mm: 3.1%	
de Muniz, 1986 [45]	1554	12–13	≥6 mm A: 9.9%. B: 2.9% ≥9 mm A:4.2% B: 2.4%		2/3 overlap: A: 8.1% B: 3.8% 3/3 overlap: A-3.5%. B-2%		A: 2.1%. B: 1.9%	
Dimberg, 2015 [46]	3 Y: 457 7 Y: 386 11.5 Y: 277	3 to 7 to 11.5	4–6 mm: 3 Y: 21.1%, 7 Y: 12.3%, 11.5 Y: 14.8% >6 mm: 3 Y: 2.9%, 7 Y: 3.7%, 11.5 Y: 6.5%		>2/3: 3 Y: 5.8%, 7 Y: 2.6%, 11.5 Y: 18.4% complete with gingival trauma: 2.2% (only 11.5 Y)	3 Y: 54.9%, 7 Y: 9.6%, 11.5 Y: 0.4%		
Esa, 2001 [48]	1519	12–13	>4 mm: 41.5%	<0 mm: 3.1%		2.0%		
Fernandes, 2008 [50]	148	3–6	≥4 mm: 33.1%		≥3 mm: 34.1%	35.1%		
Ferro, 2016 [51]	380	14	>3 mm: 48% >5 mm: 15%		>3 mm: 39% >5 mm: 9%	1.4%		
Frazao, 2006 [53]	13,801	12 and 18	≥4 mm: A-28.9%–B-21.1%	≤0 mm: A-2%–B-2.2%		A-9.2%–B-8.6%		
Gàbris, 2006 [54]	483	16–18	Ant. max. OJ: 60.8%	Ant. mand. OJ: 1.8%	deep bite: 26.1%	10.8%		
Gois, 2012 [55]	212	8–11	1–3 mm: 63.7% >3 mm: 33.5%	<1 mm: 2.8%	>1 mm: 19.3% 1–3 mm: 52.4% >3 mm: 28.3%			
Grabowski, 2007 [56]	3041 A: 4.5 Y B: 8.3 Y	4.5 and 8.3	>4–6 mm: A: 9.6%-B: 12% >6 mm: A: 3.2%–B: 4.2%	<0 mm: A: 1.3%–B: 2.7%	>2 mm: A: 33.2%-B: 46.8%	A: 11.4%–B: 9.5%		
Gudipaneni, 2018 [58]	500	7–12	>2 mm: 22.2% <1 mm: 11.4%		>2 mm: 23.4% <1 mm: 12.2%	4.6%		
Hassanali, 1993 [62]	412 A: Maassai 235 B: Kikuyu 116 C: Kalejin 61	3–16	0.5–11.5 mm: A: 84.3% B: 99.1% C: 85.2%		0.5–9.9 mm: A: 78.6% B: 9.3% C: 59.0%	0.5–8.5 mm: A: 18.3% B: 9.3% C: 24.6%		
Howell, 1993 [63]	154	13–17			10–50%: 61%	4.5%		
Ingervall, 1975 [64]	200	8–16	6–9 mm: 7%	0-(−2) mm: 1.5%	5 < 7 mm: 15% ≥7 mm: 2%		2.0%	
Jamilian, 2010 [65]	350	14–17	>9 mm: 3.1%	>−3.5 mm: 2.3%	7.7%		3.7%	
Jerez, 2014 [66]	120	3–6	>9 mm: 47.1%	3.9%	39.2%	2.0%		
Johnson, 2000 [68]	294	9.9–11.3	>6 mm: 17%	≥1 mm: 3.4%		4.0%		
Kabue, 1995 [69]	221	3–6	13%		deep: 13%		12.0%	
Kalbassi, 2019 [70]	1208	7–15	increased: 20.1%	9.8%	>4 mm: 17.8%	8.4%		6 ≥ 5 mm: 6%
Kasparviciene, 2014 [71]	709	5–7	edge–edge: 9.3% 0–2 mm: 40.8% >2 mm: 46.1%	<0 mm: 3.8%	edge–edge: 9% 1–3 mm: 57.4% >3 mm: 31.0%	2.6%		3.0%
Komazaki, 2012 [74]	963	12–15	>6 mm: 9.8%	<−1 mm: 1.2%	>5 mm: 8.9%	<−4 mm: 0.5%		
Lux, 2009 [78]	494 M: 237 F: 257	8.6- 9.6	2–3 mm: M: 24.7%–F: 29.1% 3–4 mm: M: 23.4%–F: 22.8% 6–9 mm: M: 6%-F: 4.7%		3–4 mm: M: 21.7%-F: 25.3% 4–5 mm: M: 20.9%–F: 16.5% 5–6 mm: M: 10.6%–F: 3.1% 6–7 mm: M: 0.9%–F: 0.8% >7 mm: M: 2.1%–F: 1.2%	3.0%–F: 4.3%		
Madiraju, 2021 [79]			>3.5 mm: 28.4%		>2/3 overlap: 16.3%		6.0%	
Mail, 2015 [80]	50	12	>2 mm: 98%	6.0%		4.0%		
Martins, 2009 [81]	264	10–12	0.1–2 mm: 3.4% 2–3 mm: 33.7% >3 mm: 50% edge–edge: 3.8%		0.1–2 mm: 19.7% 2–3 mm: 30.3% >3 mm: 36.7% edge–edge: 4.2%		9.1%	0.6%
Martins, 2019 [82]	1612	11–14	≤4 mm: 94.8% >4 mm: 5.2%	4.9%			≤2 mm: 99.2% >2 mm: 0.7%	
Mohamed, 2014 [84]	106	8–10	>6 mm: 17.8% total increased: 42.5%	4.7%	increased: 55.7% palatal trauma: 0.9%	0.9%		
Mtaya, 2009 [85]	1601	12–14	1–4.9 mm: 73.3% 5–8.9 mm: 11.1% ≥9 mm: 0.4%	0–1.9 mm: 8.2% ≥2 mm: 0.2%.	0.1–2.9 mm: 65.9% 3–4.9 mm: 17.9% ≥5 mm: 0.9%		0–1.9 mm: 8.9% ≥2 mm: 6.1%;	
Mtaya, 2017 [86]	253	3–5	1–4.9 mm: 65.6% 5–8.9 mm: 1.2%	<0–1.9 mm: 5.5%	0.1–2.9 mm: 60.9% 3–4.9 mm: 6.3%	0–1.9 mm: 15.8% ≥2 mm: 2.8%		
Murshid, 2010 [87]	1024	13–15	4–6 mm: 24% >6 mm: 5%		4–6 mm: 27% >6 mm: 13%			
Muyasa, 2012 [88]	1382	12–15	≥4 mm: 36.4%			14.0%		
Ng’ang’a, 1991 [89]	251	13–15	>4 mm: 23.1%		>2/3 overlap: 7.6%	9.6%		
Ng’ang’a, 1996 [90]	919	13–15	≥6 mm: 10%	0.0%	≥5 mm: 7%		8.0%	
Nguyen, 2014 [92]	200	12 and 18	>3.5 mm: 36.3%		>3.5 mm: 26.3%			
Onyeaso, 2004 [95]	636	12–17	>3 mm: 15.7%		>middle third: 14.1%	7.1%		
Oshagh, 2010 [96]	700	0–14	large: 30%	18.0%	deep bite: 53%	11.0%		
Perillo, 2010 [98]	703	12.2 ± 0.6	>4 mm: 16.2% 0–4 mm: 83.2%	<0 mm: 0.6%	>4 mm: 20.2% 0–4 mm: 79.2%	0.7%		
Perinetti, 2008 [99]	1198	7–11	>3 mm: 45%		>middle third: 38.1%			
Pineda, 2011 [100]	307	6–11	>6 mm: 18.9%		with gingival/palatal trauma: 11.6%	1.7%		
Rapeepattana, 2019 [101]	202	8–9	0–3.5 mm: 46.7% 3.5–6 mm with comp lips: 40.5% 3.5–6 mm with incomp.lips: 2.6% 6.0–9.0 mm: 3.1% >9 mm: 1.5%	5.6%	0–3.5 mm: 50.3% >3.5 mm without gingival contact: 20.5% >3.5 mm with gingival contact: 21.0% >3.5 mm with gingival trauma: 6.7%	1.5%		
Rauten, 2016 [102]	147 A (6 Y): 69 B: (9 Y): 78	6 and 9	>3 mm: A: 10.1%–B: 55.1%		>1/3 overlap: A: 7.2%–B: 47.4%	A: 17.39%–B: 11.53%		
Robke, 2007 [103]	434	2–6	>3 mm: 30.6%	2.3%	>3 mm: 16.1%	14.7%		
Rwakatema, 2007 [106]	289	12–15	>4 mm: 12.1%	>0 mm: 0.3%		6.2%		
Sanadhya, 2014 [107]	947	12–15	0 mm: 1.4% 1 mm: 36.1% 2–3 mm: 49% ≥4 mm: 12.7%	0 mm: 97.9% ≥ 1 mm: 2.1%		0 mm: 97.7% ≥1 mm: 2.3%		
Sánchez-Pérez, 2013 [108]	249	15	>2 mm: 39%	0.3%		4.5%		
Sepp, 2017 [111]	392	7.1–10.4	≥3.5 mm: 37.5%	1.0%	≥3.5 mm: 51.8%			
Sepp, 2019 [112]	390	4–5	≥3.5 mm: 15.6%	2.3%	≥3.5 mm: 38.7%	3.1%		
Shalish, 2013 [113]	432	7–11	≥7 mm: 3.7%	5.2% (impinging)	6.5%			
Singh, 2011 [114]	927	12	0–2 mm: 88.3% >2 mm: 11.7%	0–2 mm: 97.8% >2 mm: 2.1%			0 mm: 98.2% ≥1 mm: 1.8%	
Sonnesen, 1998 [116]	104	7–13	≥6 mm: 36.5%	1.9%	≥5 mm: 30.8%	3.8%		
Stahl, 2003 [117]	8864 A: Deciduous dentition B: Mixed dentition	2 > 10	A > 3 mm: 16.8% B >4 mm: 13.8%	A: 1.1% B: 1.2%	>middle third A: 1.1% B: 1.2%	A: 6.7% B: 2.8%		
Steinmassl, 2017 [119]	157	8–10	1 mm: 7.0% 2 mm: 15.9% 3 mm: 27.4% 4 mm: 19.1% 5 mm: 15.9% 6 mm: 9.6% 7 mm: 1.9% 10 mm: 0.6%	0 mm: 0.6% −1 mm: 0.6% −2 mm: 0.6% −4 mm: 0.6%	0 mm: 1.9% 1 mm: 4.5% 2 mm: 15.3% 3 mm: 27.4% 4 mm: 22.3% 5 mm: 17.8% 6 mm: 8.3% 7 mm: 2.6%			
Sundareswaran, 2019 [120]	1554	13–15	>3 mm: 11.8% edge–edge: 5.5%	1.6%	>1/2 overlap: 27.5%	1.6%		
Sunil, 2019 [121]	100	13–17	>3 mm: 26%		>2 mm: 17%			
Tausche, 2004 [123]	1975	6–8	>0 ≤ 3.5 mm: 60.2% >3.5 ≤ 6 mm: 25.3% >6 ≤ 9 mm: 5.0% >9 mm: 1.1%	<−1 mm: 0.5% <0 ≥ −1 mm: 0.9%	<3.5 mm: 53.8% ≥3.5 mm without gingival contact: 15.8% complete without trauma: 15.9% complete with trauma: 14.5%		NONE: 82.3% 1–3 mm: 14.9% 4–6 mm: 2.4% >6 mm: 0.4%	
Thilander, 2001 [124]	4724	5–17	>4 mm: 25.8%	5.8%	>4 mm: 21.6%	9.0%		
Todor, 2019 [126]	960	7–14			>1/3 overlap/28.7%		7.9%	
Uematsu, 2012 [127]	2378 A: 12–13 B: 15–16	12–13 15–16	>6 mm: A: 9.4%-B: 7.8%		deep: A: 8.4%–B: 5.8%	A: 0.6%–B: 1.2%		
Wagner, 2015 [130]	377	3	≥3 mm: 41.2%				10.9%	
Yu, 2019 [132]	2810	7–9	>3 ≤ 5 mm: 23.5% >5 ≤ 8 mm: 12.1% >8 mm: 5.2%		>2/3 overlap: 6.2%	4.3%		
Zhou, 2017 [133]	2335	3–5	>3 ≤ 5 mm: 26% >5 ≤ 8 mm: 6.9% >8 mm: 0.9%		>1/2 ≤ 3/4: 22.3% >3/4 < 1: 26.2% all cover: 15.3%			

Legend: Prevalence of overjet, reversed overjet, overbite, and open bite are noted as in the included article. Y: age range is noted, but if not available, the mean ± SD are noted and * if SD not mentioned in article. Only mandatory if the groups mentioned are under subjects. Abbreviations: Y: years, SD: standard deviation, Y:years, M: months, ant.: anterior, max.: maxillary, mand.: mandibular, incomp.: incompetent.

#### 3.3.3. Transversal Occlusion

The type of crossbite was not specified in 12 studies, and 58 investigated at least one type of crossbite. The mean prevalence of a non-specified crossbite in the studied populations was 6.2 ± 7.8% (range 1.0–36.0%). Additionally, 7.6 ± 6.0% presented a posterior crossbite (range 0.3–32.0%), 8.3 ± 2.9% (range 4.0–13.5%) presented a unilateral crossbite, and 2.5 ± 1.8% (range 0.0–6.5%) presented a bilateral crossbite. Nine studies dealt with the prevalence of scissor bite, reporting a weighted mean prevalence of 2.2 ± 3.4% (range 0.0–14.3%). The presence of a forced bite (crossbite with lateral or frontal shift) was assessed in nine studies and was found in 13.7 ± 7.7% of the included population (range 1.1–22.5%).

#### 3.3.4. Tooth Anomalies

Hypodontia (wisdom teeth excluded) was reported in 44 articles, with a mean reported prevalence of 6.5 ± 4.2% (range: 0.0–18.6%). Hyperdontia was reported with a mean prevalence of 2.1 ± 1.2% (range: 0.2–4.5%) in 19 studies, and mesiodens showed a weighted mean prevalence of 1.3 ± 0.5% (range: 0.3–1.6%). In all of these studies, X-rays were taken. The prevalence of hypo-hyperdontia—the simultaneous occurrence of both abnormalities in the same person—was 0.4 ± 0.1% (range: 0.3–0.5%).

Only a few studies included other dental anomalies, such as impacted teeth (12 studies), ectopic eruption (8 studies), and transposition of teeth (6 studies). The mean prevalence of impacted teeth, ectopic eruption, and transposition was found in 4.0 ± 2.4% (range: 0.5–12.9%), 5.3 ± 3.5% (range: 0.9–11.1%), and 0.9 ± 0.6% (range: 0.1–1.4%), respectively.

#### 3.3.5. Space Anomalies 

Crowding was not defined in the vast majority if the studies assessing this parameter [1,21,22,25,27,28,32,33,35,37,40,44,45,46,47,53,54,55,63,65,66,67,68,69,70,79,80,82,83,88,92,96,98,101,107,108,109,112,113,114,116,117,119,120,121,124,125,132,133]. The remaining studies used the Irregularity Index (Little, 1975) [51], the method of Björk [87,90,106], overlapping of erupted teeth due to insufficient space or lack of space for teeth to erupt in the dental arch [41,58,81,127] and others.

In general, crowding represented a mean prevalence of 33.8 ± 18.1% (range: 0.8–93.4%). When assessed separately for the maxillary and mandibular arch, a weighted mean prevalence for crowding of 20.8 ± 14.5% (range: 1.7–77.9%) and 19.7 ± 15.8% (range: 0.3–83.3%) was found, respectively. The mean prevalence of spacing was reported in 18.7 ± 13.7% of the samples (range: 1.2–59.5%) and demonstrated 23.4 ± 20.1% (range: 1.8–62.2%) and 12.8 ± 10.6% (range: 1.3–30.0%) prevalence in the upper and lower jaw, respectively. The weighted mean prevalence of a midline diastema was reported in 13.8 ± 14.2% (range: 1.0–73.0%).

#### 3.3.6. Oral Habits

A total of 11 articles reported oral habits, with some of them focusing on changes over time, while others just mentioned oral habits in correlation with malocclusion. The prevalence of oral habits ranged from 10.9% to 40.2%. Further details can be found in Table 3.

#### 3.3.7. Geographic Differences

The prevalence of malocclusion and of the studied occlusal traits on the different continents is presented in Table 4, Table 5, Table 6 and Table 7 For this, the studies were clustered per continent as follows: Africa, America, Asia, Europe, and Oceania.

### 3.4. Risk of Bias

The risk of bias of the included articles determined according to the MINORS tool is shown in Table 8. The scores of each article are plotted in Figure 2 and Figure 3 for non-comparative and comparative studies, respectively, and are sorted by publication year, from oldest to newest. The lowest score for non-comparative studies was 2, and the highest was 10, with a possible maximum score of 16. For comparative studies, the lowest score was 5, and the highest was 13, with a possible maximum of 24. A very discrete tendency to better article quality over time can be found in both comparative and non-comparative studies.

Risk of bias assessment of the 90 non-comparative studies according to the MINORS tool.

Risk of bias assessment of the 33 comparative studies according to the MINORS tool.

## 4. Discussion

This systematic review was performed to identify, synthesize, and assess the available evidence on the prevalence of malocclusion and other orthodontic features in subjects younger than 18 years old.

According to the WHO, before an epidemiological survey can be carried out, the investigators need to decide the following: whether to perform it at a local, regional, or national level; what variables to examine; which age groups to include [134]. Prior to the start, clear definitions should be provided to the study variables and measurement protocols and how to record the results should be defined. Ethnicity and geographical data are also indispensable [134], and performing a prospective calculation of the sample size and eventual subsamples is advised [10], since diagnostic criteria need to be based on comparable data in a representative sample. When reporting the results, all of the materials and methods should be described in detail to be able to evaluate possible selection and/or design bias.

Sample size is an important factor. Only 32 of the 123 studies included in this systematic review reported sample size estimation prior to the start. Size differences ranging from 50 to 13.801 individuals can be found in the included studies, which can partially explain the large ranges found in the prevalence of some of the studied malocclusion traits. The use of patient samples can also introduce additional bias over random samples since patients seek dental or orthodontic treatment for a reason. In this sense, it is preferable to conduct an epidemiological study on a population-based sample rather than on patient populations.

It is hard to draw solid conclusions regarding different orthodontic parameters due to the large variety of methods used to assess the different orthodontic features. Some examples of this inconsistency can be found in the description of overjet. The included studies defined increased overjet as >2.5 mm [29], >3 mm [81], >4 mm [14], and >6 mm [22], which makes it impossible to compare the data. Due to this heterogeneity in reporting, it was impossible to distinguish prevalence of occlusion according to age or dental stage, since most articles report groups with a large age range and do not provide this distinction.

The Dental Aesthetic Index (DAI) was used to report the findings of several studies, which is in accordance with the methods recommended by the WHO to standardize epidemiological data on malocclusion and treatment need [134]. However, the DAI is not a complete measure of malocclusion, but rather an aesthetic treatment need index since it does not measure occlusal parameters such as crossbite, asymmetry, midline deviation, missing molars, or impacted teeth [114].

Other studies used the Dental Health Component of the Index of Orthodontic Treatment Need to assess different orthodontic features (Table 1). Araki et al. stated that only the IOTN can diagnose the type of malocclusion, such as increased or reverse overjet, overjet, deep bite, open bite, and crowding [22]. Although they score some orthodontic features, neither the IOTN nor DAI were developed to perform epidemiological surveys on the prevalence of orthodontic features, but rather to assess orthodontic treatment need [135,136]. Thirty-nine of the studies included in this Systematic Review used X-rays, ten of which were performed in schoolchildren. The British Orthodontic Society states that each radiograph must be clinically justified because the prescription of a radiograph is a procedure with a low but nevertheless inferred risk [137]. In this context, the assessment of some orthodontic features such as the presence of hypodontia, impacted, or retained teeth, etc., remains a problem since taking radiographs for epidemiological studies is not initially indicated.

Oral habits can influence the development of malocclusion [71]. Thumb and finger sucking can cause an open bite in preadolescent children, and when such oral habits are persistent, increased overjet, decreased overbite, and crossbite can be observed [138]. The use of pacifiers has been linked to an increased prevalence of an anterior open bite and posterior crossbite [139]. Furthermore, tongue thrust at swallowing or rest can cause malocclusions such as open bite [4]. Stahl et al. found a decrease in oral habits from 40.2% in deciduous dentition to 26.1% in mixed dentition [118]. The protocols to diagnose infantile swallowing, sucking habits, and tongue position are rarely mentioned in the studies and are mostly based on subjective data. Often, the assessment of a child’s current and previous oral habits is based on information obtained from the parents, either informally or through non-validated questionnaires [71]. Therefore, there is an urgent need to develop methods that allow for the objective quantification of oral habits. The geographical differences in the prevalence of malocclusion traits are also worth mentioning. For instance, the prevalence of Angle Class II malocclusion was reported to be around 25% in America, Asia, and Europe, while the mean prevalence in Africa was 8.80 ± 10.36%. The weighted mean prevalence for Class III malocclusions for Europe, America, Africa, and Asia is 3.4 ± 1.4%, 4.1 ± 1.4%, 4.8 ± 4.2%, and 7.8 ± 4.2%, respectively, which is in accordance with the conclusions of Proffit that Class III malocclusions are more prevalent in Asian populations [4]. The mean prevalence of anterior crossbite was the highest in Asia (10.3 ± 6.5%) and the lowest in America (1.0 ± 0.6%).

Regarding transversal discrepancies, while posterior crossbites were more prevalent in America (13.0 ± 1.2%) than in Africa (5.5 ± 2.8%), a forced bite was the most prevalent in Africa (14.7 ± 10.3%) followed by Europe (13.7 ± 5.5%), and a scissor bite was the most prevalent in Africa (10.3 ± 4.8%). The prevalence of tooth anomalies ranged from 3.4 ± 2.2% in Africa to 8.1 ± 6.3% in Europe for hypodontia and from 0.3 ± 0.2% in Africa to 2.7 ± 1.6% in Asia for hyperdontia.

The geographical differences found in this systematic review are in accordance with the findings reported by Cenzato et al., which suggest that genetic and environmental factors that typically influence malocclusion traits in each population [140]. However, these differences could also be accounted for by the large heterogeneity in study designs, classifications for tooth anomalies, and a lack of clear international terminology, as previously reported by Anthonappa et al. [141]. Specifically, for the articles included in this review, the large ranges reported and the disparity in the number of studies per continent could have also played a role in the observed geographical differences.

## 5. Conclusions

A plethora of methods to determine the prevalence of malocclusion and orthodontic features was found across the included studies, which makes the data regarding prevalence of malocclusion unreliable. The mean prevalence of Angle Class I, Class II and Class III malocclusion was 51.9% (SD 20.7), 23.8% (SD 14.6) and 6.5% (SD 6.5), respectively. The prevalence of anterior crossbite, posterior crossbite and crossbite with functional shift was 7.8% (SD 6.5), 9.0% (SD 7.34) and 12.2% (SD 7.8), respectively. The prevalence of hypodontia and hyperdontia were reported to be 6.8% (SD 4.2) and 1.8% (SD 1.3), respectively. For impacted teeth, ectopic eruption and transposition, a mean of 4.9% (SD 3.7), 5.4% (SD 3.8) and 0.5% (SD 0.5) was found, respectively. There is an urgent need to establish methodological protocols for epidemiological studies in orthodontics, which should be reached in consensus with academia and professional societies. Only this will allow objective data to be obtained on which recommendations to the healthcare sector and involved stakeholders can be based.

## Figures and Tables

**Figure 1 ijerph-19-07446-f001:**
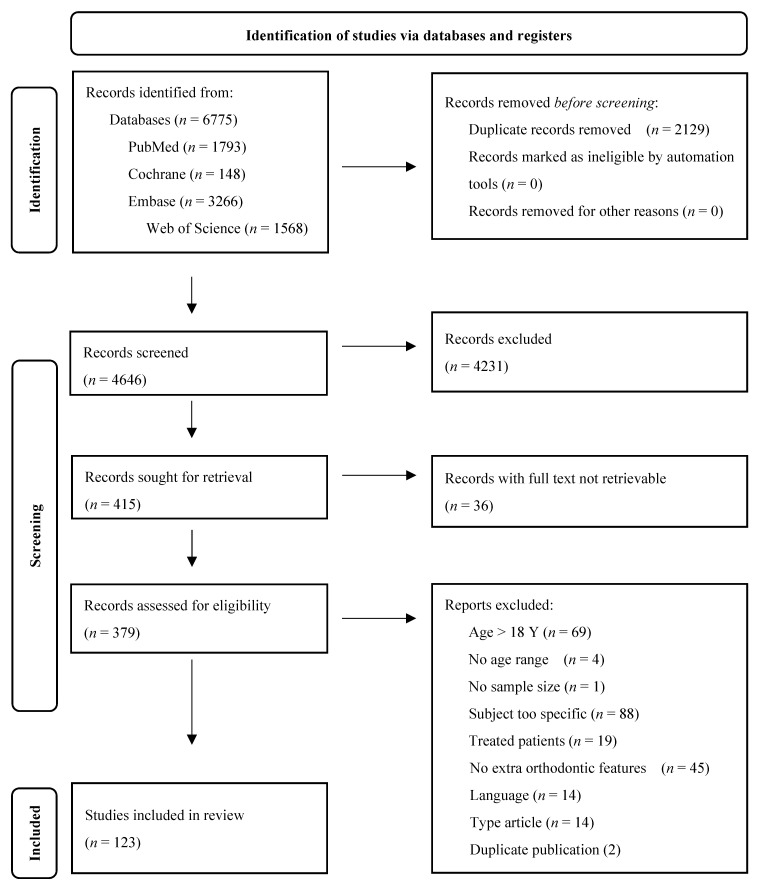
PRISMA flow diagram of the study selection process.

**Figure 2 ijerph-19-07446-f002:**
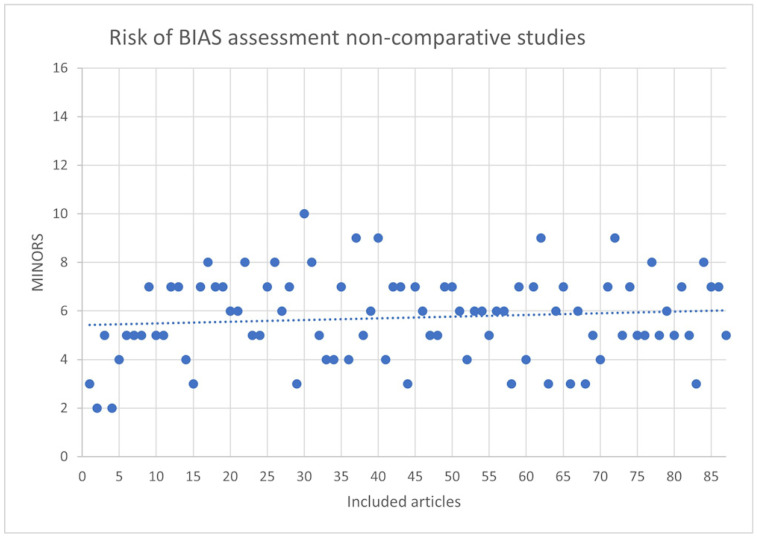
Risk of bias assessment for non-comparative studies.

**Figure 3 ijerph-19-07446-f003:**
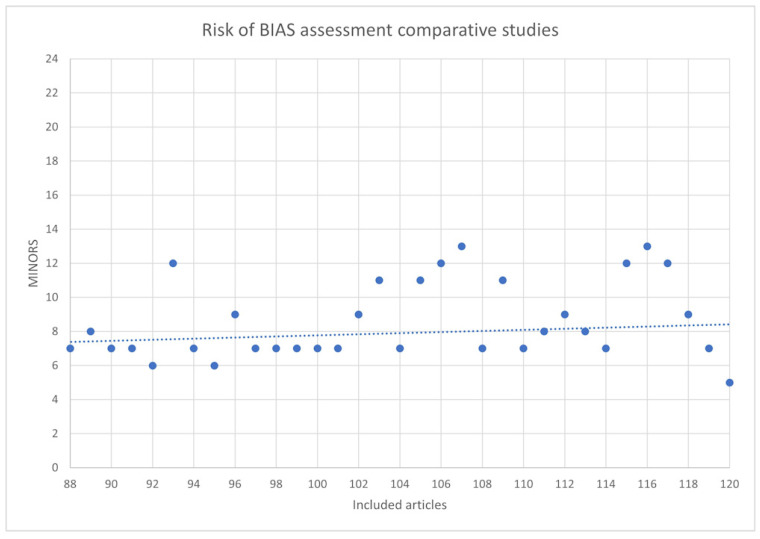
Risk of bias assessment for comparative studies.

**Table 1 ijerph-19-07446-t001:** Characteristics of and methods used in the included studies.

Author Year of Publication	Type Study	Population		Subjects					Registration					
		Country	Continent	Nr.	Age in Y	Sch. Ch./Ch.	Pat.	Pat. Rec.	Clin. Exam	X-rays OPT	Study Casts	Photographs	Interv./Quest.	Method
Aasheim, 1993 [11]	ES	Norway	Europe	1953	9	X		X		X	X	X		NM
Abu Alhaija, 2005 [12]	ES	JordanSaudi	Asia	1003	13–15	X			X	X	X			ANGLE, BJÖRK
Abumelha, 2018 [13]	CS	Arabia	Asia	526	6–12	X			X					ANGLE
Alajlan, 2019 [14]	CS	Saudi Arabia	Asia	520	7–12	X			X					ANGLE IOTN
Al-Amiri, 2013 [15]	CS	USA	America	496	16 y 3 m *			X	X	X	X			NM
Alberti, 2006 [16]	CS	Italy	Europe	1577	6–10	X			X					NM
al-Emran, 1990 [17]	ES	Saudi Arabia	Asia	500	13.5–14.5	X			X	X				BJÖRK
Alkilzy, 2007 [18]	ES	Syria	Asia	234	2–16		X		X	X	X			NM
Alsoleihat, 2014 [19]	CS	Jordan	Asia	85	14–18	X			X	X	X			NM
Altug-Atac, 2007 [20]	ES	Turkey	Asia	3043	8.5–14.75			X		X	X			NM
Arabiun, 2014 [21]	CS	Iran	Asia	1338	14–18	X			X					ANGLE
Araki, 2017 [22]	CS	Mongolia	Asia	420	10–16	X			X	X				IOTN
Baccetti, 1998 [23]	CS	Italy	Europe	5450	7–14			X		X	X			NM
Badrov, 2017 [24]	CS	Croatia	Europe	4430	6–15			X		X				NM
Baral, 2014 [25]	CS	Nepal	Asia	506	3–5	X			X					ANGLE, FOSTER & HAMILTON. DAI
Baron, 2018 [26]	CS	France	Europe	551	15.23 *			X		X		X		
Baskaradoss, 2013 [27]	CS	India	Asia	300	11–15	X			X					DAI
Behbehani, 2005 [28]	ES	Kuwait	Asia	1299	13–14	X			X		X			ANGLE
Berneburg, 2010 [29]	CS	Germany	Europe	2015	4–6	X			X					
Bhardwaj, 2011 [30]	CS	India	Asia	622	16–17	X			X					DAI
Bhayya, 2011 [31]	CS	India	Asia	1000	4–6	X			X					FOSTER & HAMILTON
Bilgic, 2015 [32]	CS	Turkey	Asia	2329	12–16	X			X					ANGLE, IOTN
Bourzgui, 2012 [33]	ES	Morocco	Africa	1000	8–12	X			X					ANGLE, BJÖRK
Calzada Bandomo, 2014 [34]	ES	Cuba	America	210	5–11	X			X					NM
Campos-Arias, 2013 [35]	ES	Costa Rica	America	88	7.0 *	X			X					ANGLE
Carvalho, 2011 [36]	CS	Brazil	America	1069	5–5 y 11 m	X			X				X	NM
Chauhan, 2013 [37]	CS	India	Asia	1188	9–12	X			X					ANGLE, DAI
Ciuffolo, 2005 [38]	ES	Italy	Europe	810	11–14	X			X					BJÖRK
Coetzee, 2000 [39]	ES	South Africa	Africa	214	3–8	X			X				X	FOSTER & HAMILTON
Cosma, 2017 [40]	ES	Romania	Europe	172	3–6	X			X					BJÖRK, FOSTER & HAMIL-TON
Dacosta, 1999 [41]	CS	Nigeria	Africa	1028	11–18	X			X					ANGLE
Daou, 2019 [42]	CS	Lebanon	Asia	334	7.31 ± 2.17		X	X	X	X				NM
de Almeida, 2008 [43]	ES	Brazil	America	344	3.94 *	X			X					FOSTER & HAMILTON
de Araújo Guimarães, 2018 [44]	CS	Brazil	America	390	8–10	X			X				X	DAI
de Muniz, 1986 [45]	ES	Argentina	America	1554	12–13	X			X					NM
Dimberg, 2015 [46]	LS	Sweden	Europe	277	3, 7 and 11.5	X			X				X	ANGLE
Endo, 2006 [47]	ES	Japan	Asia	3358	5–15		X			X	X			NM
Esa, 2001 [48]	ES	Malaysia	Asia	1519	12–13	X			X				X	DAI
Esenlik, 2009 [49]	ES	Turkey	Asia	2599	6–16					X				NM
Fernandes, 2008 [50]	ES	Brazil	America	148	3–6	X			X					NM
Ferro, 2016 [51]	CS	Italy	Europe	380	14	X			X		X			IOTN
Ferro, 2016 [52]	CS	Italy	Europe	1960	3–5	X			X					ANGLE
Frazao, 2006 [53]	ES	Brazil	America	13,801	12 and 18			X	X					DAI
Gàbris, 2006 [54]	ES	Hungary	Europe	483	16–18	X			X					ANGLE, DAI
Gois, 2012 [55]	LS	Brazil	America	212	8–11	X			X				X	ANGLE, DAI
Grabowski, 2007 [56]	CS	Germany	Europe	3041	4.5 * and 8.2 *	X			X					ANGLE
Gracco, 2017 [57]	CS	Italy	Europe	4006	9–16			X		X				NM
Gudipaneni, 2018 [58]	ES	Saudi Arabia	Asia	500	7–12	X			X					ANGLE, IOTN
Guttierez Marin, 2019 [59]	ES	Costa Rica	America	157	6–12		X			X				NM
Harris, 2008 [60]	RS	USA	America	1700	12–18			X		X				NM
Harris, 2008 [61]	RS	USA	America	1700	12–18			X		X				NM
Hassanali, 1993 [62]	ES	Kenya	Africa	412	3–16	X			X		X			NM
Howell, 1993 [63]	ES	Australia	Oceania	154	13–17		X		X					ANGLE
Ingervall, 1975 [64]	ES	Finland	Europe	200	8–16	X				X	X			ANGLE
Jamilian, 2010 [65]	ES	Iran	Asia	350	14–17	X			X					IOTN
Jerez 2014 [66]	CS	Venezuela	America	120	3–6	X			X					FOSTER & HAMILTON, ANGLE
Johannsdottir, 1997 [67]	ES	Iceland	Europe	396	6	X				X	X			BJÖRK
Johnson, 2000 [68]	ES	New Zealand	Oceania	294	9.9–11. 3	X			X					DAI
Kabue, 1995 [69]	ES	Kenya	Africa	221	3–6	X			X					FOSTER & HAMILTON, BJÖRK
Kalbassi, 2019 [70]	RS	Iran	Asia	1208	7–15	X			X		X			ANGLE, IOTN
Kasparviciene, 2014 [71]	CS	Lithuania	Europe	709	5–7	X			X					ANGLE, FOSTER & HAMILTON
Kielan-Grabowska, 2019 [72]	CS	Poland	Europe	674	6–15			X		X				NM
Kolawole, 2019 [73]	CS	Nigeria	Africa	992	1–12	X			X					DAI
Komazaki, 2012 [74]	CS	Japan	Asia	963	12–15	X			X					ANGLE, IOTN
Lagana, 2013 [75]	CS	Albania	Europe	2617	7–15	X			X				X	ANGLE, IOTN
Lagana, 2017 [76]	CS	Italy	Europe	4706	8–12	X				X				NM
Lara, 2013 [77]	CS	Brazil	America	1995	4–13			X		X				NM
Lux, 2009 [78]	ES	Germany	Europe	494	8.6–9.6	X			X					ANGLE, BJÖRK
Madiraju, 2021 [79]	CS	Saudi Arabia	Asia	282	8–9		X		X					ANGLE, IOTN
Mail, 2015 [80]	CS	Brazil	America	50	12	X			X					DAI
Martins, 2009 [81]	CS	Brazil	America	264	10–12	X			X	X		X		ANGLE
Martins, 2019 [82]	ES	Brazil	America	1612	11–14	X			X					DAI
Medina, 2012 [83]	ES	Venezuela	America	607	5–11			X		X	X	X		NM
Mohamed, 2014 [84]	CS	Malaysia	Asia	106	8–10	X			X					ANGLE, IOTN
Mtaya, 2009 [85]	ES	Tanzania	Africa	1601	12–14	X			X					ANGLE, BJÖRK
Mtaya, 2017 [86]	CS	Tanzania	Africa	253	3–5	X			X					ANGLE, BJÖRK
Murshid, 2010 [87]	CS	Saudi Arabia	Asia	1024	13–15	X			X					ANGLE, BJÖRK
Muyasa, 2012 [88]	CS	Kenya	Africa	1382	12–15	X			X				X	DAI
Ng’ang’a, 1991 [89]	ES	Kenya	Africa	251	13–15	X			X					NM
Ng’ang’a, 1996 [90]	ES	Kenya	Africa	919	13–15	X			X					ANGLE, BJÖRK
Ng’ang’a, 2001 [91]	ES	Kenya	Africa	615	8–15			X		X				NM
Nguyen, 2014 [92]	CS	Vietnam	Asia	200	12 and 18	X			X					ANGLE, IOTN
O’ Dowling, 1989 [93]	ES	Ireland	Europe	3056	7–17			X		X				NM
O’ Dowling, 1990 [94]	ES	Ireland	Europe	3056	7–17			X		X				NM
Onyeaso, 2004 [95]	ES	Nigeria	Africa	636	12–17	X			X					ANGLE
Oshagh, 2010 [96]	CS	Iran	Asia	700	0–14		X			X		X		ANGLE
Pagan- Collazo, 2014 [97]	CS	Puerto Rico	America	1911	10–14			X		X	X			NM
Perillo, 2010 [98]	ES	Italy	Europe	703	12.2 *	X			X					ANGLE
Perinetti, 2008 [99]	ES	Italy	Europe	1198	7–11	X			X				X	ANGLE
Pineda, 2011 [100]	CS	Chili	America	307	6–11		X			X				NM
Rapeepattana, 2019 [101]	CS	Thailand	Asia	202	8–9	X			X		X			ANGLE, IOTN
Rauten, 2016 [102]	ES	Romania	Europe	147	6 and 9	X			X					ANGLE, IOTN
Robke, 2007 [103]	ES	Germany	Europe	434	2–6	X			X					ANGLE
Rølling, 1980 [104]	ES	Denmark	Europe	3325	9–10	X			X	X				NM
Rozsa, 2009 [105]	ES	Hungary	Europe	4417	6–18			X		X				NM
Rwakatema, 2007 [106]	CS	Tanzania	Africa	289	12–15	X			X					DAI
Sanadhya, 2014 [107]	CS	India	Asia	947	12–15	X			X					DAI
Sánchez-Pérez, 2013 [108]	CS	Mexico	America	249	15	X			X					DAI
Seemann, 2011 [109]	CS	Germany	Europe	2975	4 and 7.8 *			X			X			NM
Sejdini, 2018 [110]	CS	Macedonia	Europe	520	7–14	X			X	X				NM
Sepp, 2017 [111]	CS	Estonia	Europe	392	7.1–10.4	X			X		X			ANGLE, ICON
Sepp, 2019 [112]	CS	Estonia	Europe	390	4–5	X			X		X		X	ANGLE, FOSTER & HAMILTON
Shalish, 2013 [113]	ES	Israel	Asia	432	7–11	X			X					NM
Singh, 2011 [114]	ES	India	Asia	927	12	X			X					DAI
Sola, 2018 [115]	CS	Spain	Europe	2500	7–11			X		X				NM
Sonnesen, 1998 [116]	CS	Denmark	Europe	104	7–13	X			X					ANGLE
Stahl, 2003 [117]	CS	Germany	Europe	8864	2–10	X			X					ANGLE
Stahl, 2003 [118]	ES	Germany	Europe	4208	6.7–13.4			X		X				NM
Steinmassl, 2017 [119]	ES	Austria	Europe	157	8–10	X			X				X	ANGLE, IOTN
Sundareswaran, 2019 [120]	CS	India	Asia	1554	13–15	X			X					ANGLE, BJÖRK
Sunil, 2019 [121]	ES	Malaysia	Asia	100	13–17	X			X					ANGLE
Swarnalatha, 2020 [122]	CS	India	Asia	1000	12–18			X		X				NM
Tausche, 2004 [123]	CS	Germany	Europe	1975	6–8	X			X			X		ANGLE, IOTN
Thilander, 2001 [124]	ES	Colombia	America	4724	5–17	X			X					ANGLE, BJÖRK
Thomaz, 2013 [125]	CS	Brazil	America	2060	12–15	X			X					ANGLE
Todor, 2019 [126]	CS	Romania	Europe	960	7–14	X			X					ANGLEBJÖRK
Uematsu, 2012 [127]	ES	Japan	Asia	2378	12–13 & 15–16	X			X					NM
Varela, 2009 [128]	ES	Spain	Europe	2108	7–16		X			X				NM
Vithanaarchchi, 2017 [129]	CS	Sri Lanka	Asia	721	8–15		X		X					NM
Wagner, 2015 [130]	CS	Germany	Europe	377	3	X			X					NM
Yassin, 2016 [131]	CS	Saudi Arabia	Asia	1252	5–12			X	X	X				NM
Yu, 2019 [132]	CS	China	Asia	2810	7–9	X			X					ANGLE
Zhou, 2017 [133]	CS	China	Asia	2335	3–5	X			X				X	FOSTER & HAMILTON

Legend: Characteristics of the included articles are provided in Table 1. Age: Age range, but if no age range was found, the mean age was noted; * Mean, if standard deviation (SD) is not mentioned in article. Abbreviations: ES: epidemiological survey; CS: cross-sectional study; LS: longitudinal study; Nr.: number of subjects; Age in Y: age range in years; Sch. Ch.: schoolchildren; Ch.: children; Pat.: patients; Pat. rec.: patient records; Clin. Exam.: clinical examination; OPT: orthopantomogram; Interv.: interviews; Quest.: questionnaires; Method reg.: method of registration; NM: Not mentioned; IOTN: Index of Orthodontic Treatment Need; DAI: Dental Aesthetic Index; ICON: Index of Complexity, Outcome and Need; ANGLE: Angle classification; BJÖRK: Björk’s method; FOSTER AND HAMILTON: method for occlusion in primary dentition.

**Table 3 ijerph-19-07446-t003:** Prevalence of oral habits.

First Author, Year	Methods																		
	Participants	Age Range in Y (Total Sample)	Location		Oral Habit in General	Non-Nutritive Sucking						Non-Nutritive Biting			Abnormal Tongue Position		Atypical Swallowing		Bruxism
	Total Number		Country			In General	Pacifier	Finger-/Thumb-Sucking	Bottle	Lip-Sucking	Lip-Inter-Position	Nail Biting	Object Biting	Cheek-/Lip-Biting	In General	Tongue Thrust	In General	Incompetent Lip-Closure	
Campos-Arias, 2013 [35]	88	7.01	Costa Rica				10.0%	19.0%	66.0%								10.2%		
Coetzee, 2000 [39]	214	3–8	South Africa					12.1%				7.5%		3.7%	7.0%		21.5%		
Howell, 1993 [63]	154	13–17	Australia					4.0%											
Kasparviciene, 2014 [71]	709	3–8	Lithuania					1.4%									5.4%		
Kolawole, 2019 [73]	992	1–12	Nigeria		13.1%			7.1%		1.3%		1.6%	1.4%			1.4%			1.4%
Lagana, 2013 [75]	2617	7–15	Tirana, Albania		81.0%		30.0%	10.2%			4.0%				9.6% (Low)		16.2%		
Mtaya, 2017 [86]	253	3–5	Tanzania			28.0%		20.9%											
Shalish, 2013 [113]	432	7–11	Israel		10.9%														
Stahl, 2003 [117]	8864	2 > 10	Germany		deciduous dentition (40.2%) mixed dentition (26.1%)	40.2% 26.1%									27.3% 28.1%			29.2% 40.9%	
Thomaz, 2013 [125]	2060	12–15	Brazil	Infancy Current			63.3% 1.1%	14.4% 3.5%				/60.3%	/55.2%	/46.1%					
Wagner, 2015 [130]	377	3	Germany				80.6%	4.3%											

Legend: The prevalence of different oral habits is noted as provided in the included articles. Age: age range in years (Y) is noted. Abbreviations: Y: years.

**Table 4 ijerph-19-07446-t004:** Prevalence of angle classification and deciduous molar occlusion according to geographical location.

Continent	Class I	Class I Mal-occlusion	Class II	Class II, 1	Class II, 2	Class III	FTP	DS	MS
Africa	58.1 ± 33.9%	71 ± 16.5%	9.7 ± 8.6%	5.8 ± 5.2%	1.4 ± 0.0%	4.8 ± 4.2%	35.9 ± 17.4%	0.9 ± 1.0%	54.8 ± 11.0%
America	13.9 ± 4.8%	50.6 ± 3.2%	28.4 ± 11.7%	17 ± 0.0% *	5.3 ± 0.0% *	13.9 ± 15.8%	73.9 ± 17.6%	7.9 ± 3.0%	15.9 ± 16.7%
Asia	50.6 ± 26.9%	41.5 ± 18.5%	27.4 ± 14.9%	19.5 ± 15.2%	4.2 ± 1.9%	7.8 ± 4.2%	41.6 ± 6.7%	10.2 ± 1.4%	36.4 ± 1.5%
Europe	47.4 ± 17.7%	46.8 ± 6.9%	25.1 ± 8.6%	16.1 ± 5.7%	4.9 ± 2.6%	3.4 ± 2.6%	28.1 ± 14.7%	24.9 ± 8.8%	47.6 ± 4.7%
Oceania	65.0 ± 0.0% *	NA	NA	15.0 ± 0.0% *	12.0 ± 0.0% *	7.0 ± 0.0% *	NA	NA	NA

Legend: The weighted mean and weighted standard deviation of the prevalence of the angle classification and deciduous molar occlusion in noted in %. * If only one study is available. NA (not available): if no data available for the given continent. Abbreviations: Class I: Angle Class I normal molar occlusion (well-aligned dental arches without any anomalies); Class I malocclusion: Angle Class I molar occlusion but with an anomaly; Class II: Angle Class II malocclusion; Class II, 1: Angle Class II, 1 malocclusion; Class II, 2: Angle Class II,2 malocclusion; Class III: Angle Class III malocclusion, FTP: flush distal plane second deciduous molars; DS: distal step second deciduous molars; MS: mesial step second deciduous.

**Table 5 ijerph-19-07446-t005:** Prevalence of different transversal malocclusions and anterior crossbite according to geographical location.

Continent	Crossbite (Not Specified)	Posterior Crossbite (Not Specified)	Posterior Crossbite Unilateral	Posterior Crossbite Bilateral	Anterior Crossbite	Scissor Bite	Forced Bite/Crossbite with Frontal/Lateral Shift
Africa	1.2 ± 0.0% *	5.5 ± 2.8%	5.5 ± 0.0% *	1.6 ± 0.0% *	5.5 ± 1.9%	10.3 ± 4.8%	14.7 ± 10.3%
America	NA	9.3 ± 6.3%	13.0 ± 1.2%	3.8 ± 1.4%	4.9 ± 3.9%	1.0 ± 0.6%	NA
Asia	8.9 ± 14.0%	6.6 ± 7.0%	5.0 ± 2.1%	5.0 ± 1.0%	10.3 ± 6.5%	1.8 ± 1.6%	11.9 ± 4.8%
Europe	5.1 ± 2.9%	8.9 ± 4.3%	8.6 ± 1.8%	1.6 ± 1.1%	5.6 ± 4.0%	1.0 ± 1.5%	13.7 ± 5.5%
Oceania	NA	NA	13.0 ± 0.0% *	6.5 ± 0.0% *	12 ± 0.0%	NA	NA

Legend: The weighted mean and weighted standard deviation of the prevalence of different transversal malocclusions: crossbite (not specified, posterior crossbite, unilateral- and bilateral crossbite, anterior crossbite, scissor bite, and crossbite with functional shift) according to geographical location are noted in %. * If only one study is available. NA (not available): if no data available for the given continent.

**Table 6 ijerph-19-07446-t006:** Prevalence of tooth anomalies according to geographical location.

Continent	Agenesis/Hypodontia	Mesiodens	Supernumerary Teeth/Hyperdontia	Hypo-Hyperdontia	Impacted/Retained Teeth (Impeded Eruption)	Ectopic Eruption	Transposition
Africa	3.4 ± 2.2%	NA	0.3 ± 0.2%	NA	3.0 ± 0.0% *	9.7 ± 0.0% *	0.2 ± 0.1%
America	5.0 ± 3.3%	1.5 ± 0.0% *	1.9 ± 0.4%	NA	3.9 ± 2.9%	1.5 ± 0.0% *	NA
Asia	8.1 ± 6.3%	NA	2.7 ± 1.6%	NA	4.8 ± 4.1%	6.0 ± 4.0%	0.5 ± 0.4%
Europe	6.9 ± 3.2%	1.3 ± 0.9%	2.3 ± 1.3%	0.4 ± 0.1%	3.8 ± 0.8%	7.5 ± 0.0% *	1.3 ± 0.7%
Oceania	7.0 ± 0.0% *	NA	1.0 ± 0.0% *	NA	5.0 ± 0.0% *	NA	NA

Legend: The weighted mean and weighted standard deviation of the prevalence of tooth anomalies: hypodontia, hyperdontia, hypo-hyperdontia, impacted/retained teeth, ectopic eruption, and transposition, according to geographical location are provided in percentages. * If only one study is available. NA (not available): if no data available for the given continent.

**Table 7 ijerph-19-07446-t007:** Prevalence of space anomalies according to geographical location.

Continent	Crowding Maxillary Arch	Crowding Mandibular Arch	Crowding	Spacing Maxillary Arch	Spacing Mandibular Arch	Spacing	Midline Diastema
Africa	23.8 ± 11.8%	24.8 ± 10.6%	24.5 ± 15.9%	32.2 ± 14.4%	22.0 ± 8.5%	32.6 ± 10.7%	36.8 ± 0.0% *
America	17.3 ± 4.3%	12.3 ± 2.7%	42.1 ± 7.3%	1.8 ± 0.0% *	1.3 ± 0.0% *	23.5 ± 4.7%	11.1 ± 7.3%
Asia	35.3 ± 21.3%	35.4 ± 23.7%	40.4 ± 22.2%	24.9 ± 17.2%	10.7 ± 5.9%	16.7 ± 14.3%	8.3 ± 4.8%
Europe	15.6 ± 19.0%	23.3 ± 19.4%	28.1 ± 11.2%	44.0 ± 15.7%	14.4 ± 2.5%	7.2 ± 13.5%	30.9 ± 20.9%
Oceania	6.0 ± 0.0% *	NA	77.4 ± 3.9%	NA	NA	45.1 ± 20.0%	NA

Legend: The weighted mean and weighted standard deviation of the prevalence of space anomalies: crowding, spacing, and midline diastema, according to geographical location given in %. * If only one study is available. NA (not available): if no data available for the given continent.

**Table 8 ijerph-19-07446-t008:** Risk of bias assessment according to the MINORS tool.

	Author, Year	M1	M2	M3	M4	M5	M6	M7	M8	M9	M10	M11	M12	T
1	Rolling, 1980 [104]	1	0	0	1	1	0	0	0	NC	NC	NC	NC	3
2	O’Dowling, 1989 [93]	1	0	0	1	1	0	0	0	NC	NC	NC	NC	2
3	Al-Emran, 1990 [17]	2	0	0	2	1	0	0	0	NC	NC	NC	NC	5
4	O’Dowling, 1990 [94]	1	0	0	1	1	0	0	0	NC	NC	NC	NC	2
5	Ng’ang’a, 1991 [89]	2	0	2	1	1	0	0	0	NC	NC	NC	NC	4
6	Aasheim, 1993 [11]	2	0	1	2	1	0	0	0	NC	NC	NC	NC	5
7	Howell, 1993 [63]	1	0	1	1	1	0	0	0	NC	NC	NC	NC	5
8	Kabue, 1995 [69]	2	0	1	1	1	0	0	0	NC	NC	NC	NC	5
9	Ng’ang’a, 1996 [90]	2	0	2	2	1	0	0	0	NC	NC	NC	NC	7
10	Johannsdottir, 1997 [67]	2	0	1	1	1	0	0	0	NC	NC	NC	NC	5
11	Sonnesen, 1998 [116]	2	0	0	2	1	0	0	0	NC	NC	NC	NC	5
12	Coetzee, 2000 [39]	2	0	2	2	1	0	0	0	NC	NC	NC	NC	7
13	Johnson, 2000 [68]	2	0	2	2	1	0	0	0	NC	NC	NC	NC	7
14	Ng’ang’a, 2001 [91]	2	0	0	1	1	0	0	0	NC	NC	NC	NC	4
15	Stahl, 2003 [118]	1	0	0	1	1	0	0	0	NC	NC	NC	NC	3
16	Onyeaso, 2004 [95]	2	0	2	2	1	0	0	0	NC	NC	NC	NC	7
17	Abu Alhaija, 2005 [12]	2	0	1	2	1	1	1	0	NC	NC	NC	NC	8
18	Behbehani, 2005 [28]	1	0	2	1	2	0	0	1	NC	NC	NC	NC	7
19	Alberti, 2006 [16]	2	0	2	2	1	0	0	0	NC	NC	NC	NC	7
20	Frazao, 2006 [53]	2	0	2	1	1	0	0	0	NC	NC	NC	NC	6
21	Gàbris, 2006 [54]	1	0	2	2	1	0	0	0	NC	NC	NC	NC	6
22	Alkilzy, 2007 [18]	2	0	2	2	2	0	0	0	NC	NC	NC	NC	8
23	Altug-Atac, 2007 [20]	2	0	0	2	1	0	0	0	NC	NC	NC	NC	5
24	Graboswki, 2007 [56]	2	0	1	1	1	0	0	0	NC	NC	NC	NC	5
25	Rwakatema, 2007 [106]	2	0	2	2	1	0	0	0	NC	NC	NC	NC	7
26	de Almeida, 2008 [43]	2	0	2	1	1	0	0	2	NC	NC	NC	NC	8
27	Fernandes, 2008 [50]	2	0	1	1	1	0	0	0	NC	NC	NC	NC	6
28	Perinetti, 2008 [99]	2	0	2	2	1	0	0	0	NC	NC	NC	NC	7
29	Robke, 2008 [103]	1	0	0	1	1	0	0	0	NC	NC	NC	NC	3
30	Martins, 2009 [81]	2	0	2	2	2	0	0	2	NC	NC	NC	NC	10
31	Lux, 2009 [78]	2	0	2	2	2	0	0	0	NC	NC	NC	NC	8
32	Rozsa, 2009 [105]	2	0	0	2	1	0	0	0	NC	NC	NC	NC	5
33	Varela, 2009 [128]	2	0	0	1	1	0	0	0	NC	NC	NC	NC	4
24	Jamilian, 2010 [65]	2	0	2	1	1	0	0	0	NC	NC	NC	NC	4
35	Murshid, 2010 [87]	2	0	2	2	1	0	0	0	NC	NC	NC	NC	7
36	Oshagh, 2010 [96]	2	0	0	1	1	0	0	0	NC	NC	NC	NC	4
37	Perillo, 2010 [98]	2	0	2	2	1	0	0	2	NC	NC	NC	NC	9
38	Bhardwaj, 2011 [30]	2	0	0	2	1	0	0	0	NC	NC	NC	NC	5
39	Campos-Arias, 2013 [35]	2	1	1	1	1	0	0	0	NC	NC	NC	NC	6
40	Carvalho, 2011 [36]	2	0	2	2	1	0	0	2	NC	NC	NC	NC	9
41	Pineda, 2011 [100]	2	0	0	1	1	0	0	0	NC	NC	NC	NC	4
42	Singh, 2011 [114]	2	0	2	1	1	0	0	1	NC	NC	NC	NC	7
43	Bourzgui, 2012 [33]	2	0	2	2	1	0	0	0	NC	NC	NC	NC	7
44	Medina, 2012 [83]	2	0	0	1	1	0	0	0	NC	NC	NC	NC	3
45	Muyasa, 2012 [88]	2	0	2	2	1	0	0	0	NC	NC	NC	NC	7
46	Uematsu, 2012 [127]	2	0	1	1	1	0	0	0	NC	NC	NC	NC	6
47	Thomaz, 2013 [125]	1	0	1	1	0	0	1	1	NC	NC	NC	NC	5
48	Al-Amiri, 2013 [15]	2	0	0	2	1	0	0	0	NC	NC	NC	NC	5
49	Baskaradoss, 2013 [27]	2	0	2	1	1	0	0	1	NC	NC	NC	NC	7
50	Chauhan, 2013 [37]	2	0	2	2	1	0	0	0	NC	NC	NC	NC	7
51	Lagana, 2013 [75]	2	0	1	1	1	0	0	1	NC	NC	NC	NC	6
52	Lara, 2013 [77]	2	0	0	1	1	0	0	0	NC	NC	NC	NC	4
53	Sánchez-Pérez, 2013 [108]	2	0	1	1	1	0	0	1	NC	NC	NC	NC	6
54	Shalish, 2013 [115]	2	0	1	1	1	0	0	1	NC	NC	NC	NC	6
55	Alsoleihat, 2014 [19]	2	0	1	1	1	0	0	0	NC	NC	NC	NC	5
56	Baral, 2014 [25]	2	0	1	2	1	0	0	0	NC	NC	NC	NC	6
57	Calzada Bandomo, 2014 [34]	2	0	1	2	1	0	0	0	NC	NC	NC	NC	6
58	Jerez, 2014 [66]	1	0	1	1	0	0	0	0	NC	NC	NC	NC	3
59	Kasparviciene, 2014 [71]	2	0	1	2	1	0	0	1	NC	NC	NC	NC	7
60	Mohamed, 2014 [84]	1	0	1	1	1	0	0	0	NC	NC	NC	NC	4
61	Nguyen, 2014 [92]	2	0	1	1	1	0	1	1	NC	NC	NC	NC	7
62	Sanadhya, 2014 [107]	2	0	2	2	1	0	0	2	NC	NC	NC	NC	9
63	Mail, 2015 [80]	1	0	1	1	0	0	0	0	NC	NC	NC	NC	3
64	Wagner, 2015 [130]	2	0	2	1	1	0	0	0	NC	NC	NC	NC	6
65	Ferro, 2016 [51]	2	0	1	2	0	0	0	2	NC	NC	NC	NC	7
66	Rauten, 2016 [102]	2	0	0	1	0	0	0	0	NC	NC	NC	NC	3
67	Araki, 2017 [22]	2	0	2	1	1	0	0	0	NC	NC	NC	NC	6
68	Badrov, 2017 [24]	1	0	0	1	1	0	0	0	NC	NC	NC	NC	3
69	Cosma, 2017 [40]	2	0	0	2	1	0	0	0	NC	NC	NC	NC	5
70	Gracco, 2017 [57]	2	0	0	2	0	0	0	0	NC	NC	NC	NC	4
71	Sepp, 2017 [111]	2	0	1	1	1	0	0	1	NC	NC	NC	NC	7
72	Steinmassl, 2017 [119]	2	0	2	2	1	0	0	2	NC	NC	NC	NC	9
73	Vitanaarchchi, 2017 [129]	2	0	1	1	1	0	0	0	NC	NC	NC	NC	5
74	Zhou, 2017 [133]	2	0	1	1	1	0	1	1	NC	NC	NC	NC	7
75	Abumelha, 2018 [13]	2	0	0	2	1	0	0	0	NC	NC	NC	NC	5
76	Baron, 2018 [26]	2	0	0	2	1	0	0	0	NC	NC	NC	NC	5
77	de Araújo Guimarães, 2018 [44]	2	0	2	1	1	0	0	2	NC	NC	NC	NC	8
78	Guttierez Marin, 2019 [59]	2	0	0	2	1	0	0	0	NC	NC	NC	NC	5
79	Mtaya, 2017 [86]	2	0	2	2	1	0	0	0	NC	NC	NC	NC	7
80	Sejdini, 2018 [110]	2	0	1	1	1	0	0	0	NC	NC	NC	NC	6
81	Sola, 2018 [115]	2	0	0	2	1	0	0	0	NC	NC	NC	NC	5
82	Alajlan, 2019 [14]	2	0	1	1	1	0	0	0	NC	NC	NC	NC	5
83	Daou, 2019 [42]	2	0	2	2	1	0	0	0	NC	NC	NC	NC	7
84	Kalbassi, 2019 [70]	2	0	0	2	1	0	0	0	NC	NC	NC	NC	5
85	Kielan-Grabowska, 2019 [72]	2	0	0	1	0	0	0	0	NC	NC	NC	NC	3
86	Rapeepattana, 2019 [101]	2	0	2	1	1	0	0	2	NC	NC	NC	NC	8
87	Sepp, 2019 [112]	2	0	1	1	1	0	0	1	NC	NC	NC	NC	7
88	Todor, 2019 [126]	2	0	2	2	1	0	0	0	NC	NC	NC	NC	7
89	Yu, 2019 [132]	2	0	1	1	1	0	0	0	NC	NC	NC	NC	5
90	Madiruja, 2021 [79]	2	0	1	1	1	0	1	2	NC	NC	NC	NC	8
91	Ingervall, 1975 [64]	2	0	1	2	1	0	0	0	0	0	0	1	7
92	de Muniz, 1986 [45]	2	0	2	1	1	0	0	0	0	1	0	1	8
93	Hassanali, 1993 [62]	2	0	2	2	1	0	0	0	0	0	0	0	7
94	Bacetti, 1998 [23]	2	0	0	2	1	0	0	0	1	0	0	1	7
95	Dacosta, 1999 [41]	2	0	0	1	1	0	0	0	0	1	0	1	6
96	Esa, 2001 [48]	2	0	2	1	1	0	0	2	1	1	0	2	12
97	Thilander, 2001 [124]	2	0	2	2	1	0	0	0	0	0	0	0	7
98	Stahl, 2003 [117]	1	0	1	1	1	0	0	0	0	0	0	2	6
99	Tausche, 2004 [123]	2	1	0	1	1	0	0	2	0	0	0	2	9
100	Ciuffolo, 2005 [38]	2	0	0	2	1	0	0	0	0	0	0	2	7
101	Endo, 2006 [47]	2	0	0	2	1	0	0	0	0	0	0	2	7
102	Esenlik, 2007 [49]	2	0	1	2	1	0	0	0	0	0	0	1	7
103	Harris, 2008 [60]	2	0	0	1	1	0	0	0	1	0	0	2	7
104	Harris, 2008 [61]	2	0	0	1	1	0	0	0	1	0	0	2	7
105	Mtaya, 2009 [85]	2	0	2	2	1	0	0	0	0	0	0	2	9
106	Berneburg, 2010 [29]	2	0	2	2	2	0	0	1	0	0	0	2	11
107	Bhayya, 2011 [31]	2	0	2	2	1	0	0	0	0	0	0	0	7
108	Seemann, 2011 [109]	2	0	2	2	1	0	0	0	0	0	0	2	11
109	Gois, 2012 [55]	2	0	2	2	1	2	1	0	0	0	0	2	12
110	Komazaki, 2012 [74]	2	1	2	2	1	0	0	1	0	1	1	2	13
111	Arabiun, 2014 [21]	2	0	2	2	1	0	0	0	0	0	0	0	7
112	Pagan-Collazo, 2014 [97]	2	0	2	2	1	0	0	2	0	0	0	2	11
113	Bilgic, 2015 [32]	2	0	0	2	1	0	0	0	0	0	0	2	7
114	Dimberg, 2015 [46]	2	0	1	1	1	0	0	0	0	1	0	2	8
115	Ferro, 2016 [51]	2	0	1	2	1	0	0	2	0	1	0	0	9
116	Yassin, 2016 [131]	2	0	1	1	1	0	0	0	0	1	0	2	8
117	Lagana,2017 [76]	2	0	0	1	1	0	0	0	0	1	1	1	7
118	Gudipaneni, 2018 [58]	2	0	2	2	1	2	0	2	0	0	0	0	12
119	Kolawole, 2019 [73]	2	0	2	2	1	0	0	2	0	1	1	2	13
120	Martins, 2019 [82]	2	0	2	2	1	0	0	2	0	1	0	2	12
121	Sundareswaran, 2019 [119]	2	0	1	1	1	0	0	1	0	1	1	1	9
122	Sunil, 2019 [120]	2	0	1	1	1	0	0	0	0	1	1	0	7
123	Swarnalatha, 2020 [121]	2	0	0	1	1	0	0	0	0	0	0	1	5

Legend: 1–87: the included non-comparative studies sorted by ascending year of publication; 88–123: the included comparative studies sorted by ascending year of publication. Abbreviations: M: MINORs item; M1: clearly stated aim; M2: inclusion of consecutive sample; M3: prospective collection of data; M4: end point appropriate to aim; M5: unbiased assessment of endpoints; M6: follow up period appropriate to aim; M7: loss to follow up less than 5%; M8: prospective calculation of study size; M9: adequate control group; M10: contemporary groups; M11: baseline equivalence; M12: adequate statistical analysis; T: total; NC: non-comparative; C: comparative studies.

## Data Availability

Data is contained within the article or Appendix A.

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
