# Peer review of "Prevalence of Orthodontic Malocclusions in Healthy Children and Adolescents: A Systematic Review"

_ijerph, 2022, doi:10.3390/ijerph19127446_

Round 1

Reviewer 1 Report

  1. There needs to be a more logical progression from the end of the Introduction to the Materials and Methods section, as currently, there is an abrupt jump. Explain what has triggered this systematic review and why it shall be of importance to the dental and wider communities.
  2. Perhaps providing a glossary of definitions for all the orthodontic terms: i.e., crowding, hypodontia, overjet, etc – this will be helpful to lay readers and those from non-dental backgrounds.
  3. The Conclusion section needs to provide more compelling take-home messages for the readership.
  4. The language, grammar, punctuation, spelling and sentence structures within the current paper, all must be thoroughly assessed and polished to ensure a succinct and coherent read. This is because there are minor discrepancies in the language, grammar, punctuation, spelling and sentence structures. Please obtain the necessary scientific English language reading and editing assistance, if need be, so that the paper has the potential to be read enjoyably by the international readership.

Author Response

We thank the reviewer for his/her comments and have answered them in the additional word file.

Answers to reviewer 1

  1. There needs to be a more logical progression from the end of the Introduction to the Materials and Methods section, as currently, there is an abrupt jump. Explain what has triggered this systematic review and why it shall be of importance to the dental and wider communities.
    • We would like to thank the reviewer for these suggestions, which we hope to have implemented in the new version of the manuscript. Regarding this first comment, we realize the last sentence of the introduction was missing and we have added it in the revised manuscript.
  1. Perhaps providing a glossary of definitions for all the orthodontic terms: i.e., crowding, hypodontia, overjet, etc – this will be helpful to lay readers and those from non-dental backgrounds.
    • We agree with the reviewer that being familiar with the orthodontic terms is crucial to understand the review. However, these terms are common knowledge to orthodontists and most dental practitioners, and due to their extensive number, would mean adding a very long appendix. Instead, we have added a reference containing this glossary of terms, for interested readers to resource to. This has been mentioned in the first paragraph of the results.
  1. The Conclusion section needs to provide more compelling take-home messages for the readership.
    • We have re-written the conclusion adding the main epidemiological data, in order to provide the reader with a global idea.
  1. The language, grammar, punctuation, spelling and sentence structures within the current paper, all must be thoroughly assessed and polished to ensure a succinct and coherent read. This is because there are minor discrepancies in the language, grammar, punctuation, spelling and sentence structures. Please obtain the necessary scientific English language reading and editing assistance, if need be, so that the paper has the potential to be read enjoyably by the international readership.
    • Following the suggestion of the reviewer, we have revised the manuscript with professional editing. The proof of editing has been attached to this revision.

Answers to reviewer 1

  1. There needs to be a more logical progression from the end of the Introduction to the Materials and Methods section, as currently, there is an abrupt jump. Explain what has triggered this systematic review and why it shall be of importance to the dental and wider communities.
    • We would like to thank the reviewer for these suggestions, which we hope to have implemented in the new version of the manuscript. Regarding this first comment, we realize the last sentence of the introduction was missing and we have added it in the revised manuscript.
  1. Perhaps providing a glossary of definitions for all the orthodontic terms: i.e., crowding, hypodontia, overjet, etc – this will be helpful to lay readers and those from non-dental backgrounds.
    • We agree with the reviewer that being familiar with the orthodontic terms is crucial to understand the review. However, these terms are common knowledge to orthodontists and most dental practitioners, and due to their extensive number, would mean adding a very long appendix. Instead, we have added a reference containing this glossary of terms, for interested readers to resource to. This has been mentioned in the first paragraph of the results.
  1. The Conclusion section needs to provide more compelling take-home messages for the readership.
    • We have re-written the conclusion adding the main epidemiological data, in order to provide the reader with a global idea.
  1. The language, grammar, punctuation, spelling and sentence structures within the current paper, all must be thoroughly assessed and polished to ensure a succinct and coherent read. This is because there are minor discrepancies in the language, grammar, punctuation, spelling and sentence structures. Please obtain the necessary scientific English language reading and editing assistance, if need be, so that the paper has the potential to be read enjoyably by the international readership.
    • Following the suggestion of the reviewer, we have revised the manuscript with professional editing. The proof of editing has been attached to this revision.

Reviewer 2 Report

The topic is interesting and really up-to-date. In my opinion a lot of orthodontic practitioners will find it useful. The paper is well organized including the following structure: abstract, introduction, main body  and conclusions. The number of references is impressive and relevant to the subject of research.

I suggest to accept the paper in present form.

Author Response

We thank the reviewer for his/her comments and have answered them in the additional word file.

  1. The topic is interesting and really up-to-date. In my opinion a lot of orthodontic practitioners will find it useful. The paper is well organized including the following structure: abstract, introduction, main body and conclusions. The number of references is impressive and relevant to the subject of research. I suggest to accept the paper in present form.
    • We thank the reviewer for his/her encouraging comments.

Reviewer 3 Report

Dear authors,

congratulations for your work. 

It seems to be methodologically well-driven.

METHODS: please provide the search strings as a supplementary material.

Did you consider syndromic patients/ the extraction of third molars /condylar fractures for your selection criteria? 

Please add as supplementary material the Prisma checklist for reporting systematic reviews.

RESULTS: you analyze the data to sort out the geographical prevalence of each malocclusion. I would suggest you to compare the prevalence of malocclusions in males vs females for each continent, as well as for socioeconomic status.

FUNDING INFORMATION, AUTHOR CONTRIBUTIONS, CONFLICT OF INTERESTS, SUPPLEMENTARY MATERIAL, ACKNOWLEDGEMENTS should be filled.

Author Response

We thank the reviewer for his/her comments and have answered them in the additional word file.

Reviewer 4 Report

The aim of the study (although not specified through the manuscript) is to identify the prevalence of malocclusion and orthodontic parameters in the general population < 18 years old.

Some major corrections need to be addressed, as specified as followed. Although I recognize the important efforts of the authors to retrieve information from 123 articles included in the analysis, it comes with no surprise the difficulty in providing an answer to the question. The research question is too broad. For example, I would have narrowed the age range considered. I do not find appropriate to provide a prevalence of Class I, II, and III while considering deciduous, mixed and permament dentition altogether. I encourage the authors to distinguish the data coming from these different population, if possible.

I would also want a better clarification of the population included (where do the participants come from? Are people seeing for orthodontic visits? This inevitable overestimate the prevalence. Are these health people? Please, see below my comments).

Abstract:

The authors should modify the conclusion of “makes it impossible to report prevalences of malocclusion”, considered that they just reported a mean prevalence in the results. I suggest softening the expression in the conclusion.

Introduction:

Providing an introduction of malocclusion as a disease of the digestive system is not appropriate for the article, as this is based solely on an orthodontic concept. I suggest providing a broad definition, such as the one provided by Angle, as historically he was the one who defined what occlusion and malocclusion were. Also, if you do not mention the orthodontic definition, in that case the following sentence on orthodontic treatment does not make any sense.

I also suggest providing a definition of the different types of malocclusion, as in the results the authors refer not only to Class I, II and III, but also to flush terminal plane, mesial and distal step.

Lines 57-60: the authors introduce the concept that other orthodontic feaures sichs as oral clefts etc are important, without justifying why they are important. This seems more a list of other prevalence, unless it is justified further.

Aim of the study is lacking at the end of the introduction.

Methods:

Lines 74-76: if cleft lip and palate are out of the scope of the review, why were they sought? Still the aim is lacking, so it is difficult to understand the objective of the search at this point.

- What do the authors mean with maxillofacial syndromes? The title of the article is “healthy children and adolescents”, therefore syndromes should be excluded. Otherwise, please specify in the eligibility criteria.

- The Intervention and the Outcome are the same.

- line 105: why “oral habits” are included in the type of examination and assessment of studies parameters? I do not think it fits in this subgroup. I suggest including the in the sample characteristics, if important.

- lines 116-131 (and also lines 256-259): authors did not delete the part of the original template

Please, move the legend to Figure 1 below the figure. Also, what does it mean “subject too specific” and “salami publication”? Also, I suggest expressing the reason of exclusion in a more formal language, e.g. “Type of article, Not healthy subjects,” etc.

I also suggest clarifying the source of the population investigated in each study. This article claims at providing epidemiological data on the prevalence of malocclusion in general population of children and adolescents. However, it is important to clarify where the samples of each study come from, as normally patients that seek treatment do have a visible malocclusion.

Moreover, how where the malocclusion calculated on the x-ray? The malocclusion included in the studies were dental, not skeletal.

Results:

lines 140-143: what does it mean children vs schoolchildren? What does it mean patients vs patient records? Please, specify.

- lines 145-149: change the parenthesis [] with () and put the % of the studies that involve that specific method, rather than the number. Put also the percentage in lines 143 where referred to the countries of the studies. I suggest moving in this paragraph the presentation of 3.3.7 Geographic differences, referring to table 4. Otherwise, the authors need to clarify in the materials that they are going to perform an analysis also on the basis of the geographical distribution. 

- lines 147: how this data can be retrieved from interviews or questionnaires?

- lines 149: spell out IOTN index.

- what is the number of studies that include a malocclusion based on permanent molars (line 168)? Please, specify.

The fact that the results are divided in sagittal and vertical should be introduced earlier in the methods.

Table 2: please, verify the information for Mohamed, 2014 [84]. Are 0.9% the same for palatal trauma and open bite? And also, what does ‘total increased’ mean?

Table 3: what does “bottle” mean?

- lines 197-198: how many studies reported information on space anomalies? How was crowding defined?

Discussion:

Paragraph lines 264-272 can be deleted or summarized. Needs of epidemiological studies are well known.

Lines 273: what does it mean the sample size? Is the sample size calculation?

Line 279: in light of this important consideration, I suggest including this data retrieved from the articles included.

Line 290: I suggest replacing “occlusal traits” with “occlusal parameters”

I suggest writing a paragraph with the limitation, where all the drawbacks identified in the discussion can be mentioned and discussed.

Conclusion:

I also encourage the authors to provide some conclusion based on the retrieved data, despite the broad range and the different methodological tools adopted from the studies. 

Author Response

(The authors gave the same response as above.)

Round 2

Reviewer 4 Report

I do not have any more comments to be addressed. I think the authors did a nice job in addressing and responding to all my concerns.